

# A wave delay neural network for solving label-constrained shortest route query on time-varying communication networks

Bing Han, Qiang Fu and Xinliang Zhang

China National Institute of Standardization, Beijing, China

## ABSTRACT

The focus of the research is on the label-constrained time-varying shortest route query problem on time-varying communication networks. To the best of our knowledge, research on this issue is still relatively limited, and similar studies have the drawbacks of low solution accuracy and slow computational speed. In this study, a wave delay neural network (WDNN) framework and corresponding algorithms is proposed to effectively solve the label-constrained time-varying shortest routing query problem. This framework accurately simulates the time-varying characteristics of the network without any training requirements. WDNN adopts a new type of wave neuron, which is independently designed and all neurons are parallelly computed on WDNN. This algorithm determines the shortest route based on the waves received by the destination neuron (node). Furthermore, the time complexity and correctness of the proposed algorithm were analyzed in detail in this study, and the performance of the algorithm was analyzed in depth by comparing it with existing algorithms on randomly generated and real networks. The research results indicate that the proposed algorithm outperforms current existing algorithms in terms of response speed and computational accuracy.

## INTRODUCTION

The shortest route query problem is a classic combinatorial optimization challenge, aiming to identify the most efficient route (minimizing cost or reducing delay) from a source node to a destination node. Solutions to this problem find extensive applications in communication networks (*Wang, Guo & Okazaki, 2009*; *Gomathi & Martin Leo Manickam, 2018*), transportation network (*Fu, Sun & Rilett, 2006*; *Neumann, 2016*), engineering control (*Nip et al., 2013*; *Lacomme et al., 2017*), and many other areas.

The shortest route query problem was initially formulated by *Dijkstra (1959)* in the 1950s. Subsequently, numerous enhanced algorithms were introduced to address this problem in time-invariant networks (*Xu et al., 2007*; *Zhang & Liu, 2009*). During that period, modifications to this problem were also proposed, including the label-constrained shortest route query on time-invariant networks (*Zhang et al., 2021*; *Likhyani & Bedathur, 2013*; *Barrett Chris, 2008*). While demonstrating certain advantages in time-invariant

Corresponding author
Qiang Fu, fuqiang@cnis.ac.cn, qiangf2023@163.com

networks, these methods still face challenges when applied to solving the shortest route query problem in time-varying networks.

The time-varying network (also known as the time-dependent network) is a dynamic network, which widely exists in the real world (*Huang, Xu & Zhu, 2022*). Compared to the traditional static networks, the time or cost of one data packet traveling an arc in the time-varying network is not constant but changes over time, which depends on the departure time from the start node and may be denoted by a piecewise function. Recently, some problems based on time-varying networks have attracted extensive attention, such as the traveling salesman problem (*Cacchiani, Contreras-Bolton & Toth, 2020*), maximum flow problem (*Zhang et al., 2018*), minimum spanning tree problem (*Huang, Fu & Liu, 2015*), project scheduling problems (*Huang & Gao, 2020*), *etc*. The shortest route query problem on time-varying networks was first studied by *Cooke & Halsey (1966)*, who proposed a Bellman-based iterative algorithm to solve the unconstrained time-varying shortest delay route problem. Since then, this kind of problem has also been studied by *Huang & Wang (2016)*, *Wu et al. (2016)*, *Huang et al. (2017)*, *Wang, Li & Tang, (2019) etc*.

Similiar to the time-invariant networks (*Feng & Korkmaz, 2013*), the shortest route query problem with constraints also exists in time-varying networks. To the best of our knowledge, the research on the constrained time-varying shortest route query problem mainly focuses on the reachiability on time-varying networks, such as delay-constrained time-varying minimum cost path problem (*Cai, Kloks & Wong, 1997*; *Veneti, Konstantopoulos & Pantziou, 2015*), and more (*Chen et al., 2022*; *Peng et al., 2020*; *Chen & Singh, 2021*; *Gong, Zeng & Chen, 2023*; *Heni, Coelho & Renaud, 2019*; *Yang & Zhou, 2017*). Choosing the appropriate path is crucial for optimizing network performance in communication networks. The constrained shortest route problem allows for the introduction of specific constraints in path selection *Ruß, Gust & Neumann (2021)*, such as bandwidth, latency, load balancing *Peng et al. (2022)*, *etc*., to meet the specific needs of the network and improve the overall efficiency of the network. Different applications and services have different requirements for network performance. The constrained shortest route problem can be used to ensure that specific quality of service standards, such as low latency and high bandwidth, are met when selecting paths in a network, thereby improving user experience. In the case of limited computing resources, the constrained shortest route problem helps to effectively manage network resources. By considering constraints, certain paths can be avoided from being too crowded, thereby improving network availability and resource utilization efficiency. However, there is limited research on the label-constrained time-varying shortest route query problem (LTSRQ).

The label-constrained shortest route query problem is of great importance in time-varying communication networks, especially in achieving efficient, reliable, and low-latency network communication. It is specifically manifested in: (1) Load balancing and resource optimization: Nodes and links in communication networks may have different performance characteristics. By considering constraints such as bandwidth and latency, path selection can be optimized to achieve load balancing, avoid overcrowding of certain paths, and improve the utilization of network resources. (2) Security: By considering label constraints, a path can be designed to ensure the security of data during transmission and

prevent security threats such as man-in-the-middle attacks. (3) Multipath transmission and traffic engineering: The label-constrained shortest route problem can be used for multipath transmission and traffic engineering, dynamically selecting the path that is most suitable for the current network state to improve the overall performance of the network.

Neural network technology has been proven to be more efficient than traditional mathematical methods in various fields (*Huang, Xu & Zhu, 2022*; *Adnéne et al., 2022*; *Zulqurnain et al., 2022*). Particularly in the investigation of the shortest route problem in time-varying networks, neural network technology, with its robust parallel computing and timing simulation capabilities, has demonstrated outstanding performance. Existing research results have substantiated the feasibility and progressive nature of neural network technology when compared to traditional mathematical methods in addressing path-related problems in time-varying networks. Therefore, in this article, a wave delay neural network (WDNN) framework is proposed to solve the LTSRQ. The purpose of LTSRQ is to find a route from the source node to the destination node having the shortest delay with a NP-hard complexity, and meet the label threshold. For example, in certain wireless broadcast networks, where the limited capacity of wireless devices necessitates selective signal reception and processing, labels are commonly employed for signal filtering. Specifically, in scenarios where the payload is associated with specific time intervals, the time required for signal processing and forwarding is generally directly proportional to the payload. As a result, such wireless broadcast networks can be categorized as labeled time-varying networks. The labeled-constrained time-varying shortest route query problem in this context aims to identify a path within the network that facilitates the transmission of signals with specific labels from the source to the destination. The proposed wave delay neural network (WDNN) is built on auto wave neurons, allowing for parallel computation. WDNN proves effective in addressing the label-constrained time-varying shortest route query (LTSRQ), arriving at the global optimal solution. Notably, unlike conventional neural networks that necessitate training, the proposed WDNN operates without any training requirements.

In general, our novelty and contributions can be summarized in the following two aspects:

- **Wave delay neural network (WDNN) framework:** A framework for Wave Delay Neural Networks (WDNN) is proposed to resolve the LTSRQ, which composed of autonomously designed and training-free wave neurons. These wave neurons are adept at handling the time-varying lengths of dynamic edges, allowing for optimal departure time selection. By assigning a state type to each neuron to restrict wave reception, the framework successfully implements label-constrained processing. Due to the adoption of parallel computation and an optimal emission time selection mechanism for neurons, this method can rapidly obtain the global optimal solution to the label-constrained time-varying shortest route query problem. It plays a crucial role in delay-sensitive communication networks.
- The effectiveness of the proposed algorithm is assessed through a thorough analysis of time complexity and a correctness proof. Performance evaluation is conducted from two

**Table 1  Explanation of symbols in WDNN.**

| Symbols | Explanation |
|---|---|
| $T_S$ | The start time of a time window. |
| $T_E$ | The end time of a time window. |
| $T_L$ | Tthe length of arc in a time window. |
| $L_n$ | The label set of a node $n$. |
| $len_e(t)$ | The length of a time-varying arc. |
| $V_P$ | The set of nodes on path $P$. |
| $E_P$ | The set of arcs on path $P$. |
| $L_P$ | The set of label of nodes on path $P$. |
| $M$ | A large integer. |
| $\alpha_i$ | The arrival time of $i$th node on the path $P$. |
| $\tau_i$ | The departure time of $i$th node on the path $P$. |
| $t$ | The current time. |
| $V_i^P$ | The set of all precursor neurons of neuron $i$. |
| $V_i^F$ | The set of all successor neurons of neuron $i$. |
| $\Delta t$ | A step (unit) of iteration. |
| $s$ | The root neuron (source node). |
| $z$ | The destination neuron (destination node). |
| $t_s$ | The earliest time from the source node is allowed. |
| $L^c$ | The constrained label set. |
| $L_i$ | The label set of neuron $i$. |
| $L_i^r$ | The recorded label set of neuron $i$. |
| $Y_{k,i}^t$ | A wave from neuron $k$ to $i$ at time $t$. |
| $P_{k,i}^t$ | The path from neuron $k$ to $i$ at time $t$. |
| $A_{k,i}^t$ | The arrival time of the wave from neuron $k$ to $i$ at time $t$. |
| $L_{k,i}^t$ | The label of the wave from neuron $k$ to $i$ at time $t$. |
| $P_i^r$ | The path recorded by neuron $i$. |
| $A_i^r$ | The set of the arrival time of each wave recorded. |
| $L_i^r$ | The label set of recorded paths. |
| $TW_{i,q}(t)$ | The time window of arc $(i, q)$ at time $t$. |

perspectives: the number of nodes and the number of time windows. The experimental results demonstrate that the proposed algorithm is capable of effectively addressing the label-constrained shortest routing query problem in time-varying networks.

To enhance the understanding of this article, Table 1 provides a summary of symbols used in the definition section and the neural network architecture design section. The rest of this article is organized as follows. 'Preliminaries' introduces the preliminary knowledge that WDNN requires. In the third section, a newly designed neural network framework, auto wave neuron, and algorithm for solving LTSRQ were proposed, and the time complexity and correctness of the proposed algorithm were analyzed, which is also the main focus of this study. Next, we conduct our experiments and evaluations in 'Experimental Results and Discussion'. Finally, the 'Conclusion' makes a conclusion of this article in brief.

## PRELIMINARIES

To ensure clarity and understanding in this study, the clear definitions will be provided for the key concepts involved. By carefully and precisely defining our concepts, it aim to ensure that our analysis is rigorous and well-informed, contributing to a comprehensive understanding of the study's foundations and findings.

**Definition 1 (Time window)** *Huang, Xu & Zhu (2022)*: A triple $(T_S, T_E, T_L)$ is defined as a time window if and only if the $T_E > T_S$, and where the $T_S$ is the start time of time window, the $T_E$ is the end time of time window, the $T_L$ is a constant number that denotes the length of arc in this time window.

**Definition 2 (Time-varying function)** *Huang, Xu & Zhu (2022)*: A piecewise function $f(t)$ is defined as a time-varying function if and only if $t$ is a time variable. If to devide the time-varying function, it can be divided into multiple time windows. That is to say, a time-varying function is a functional representation of one or more time windows.

**Definition 3 (Label node)**: A node is defined as a label node if and only if it has at least one label. The label set of a node $n$ is denoted as $L_n$.

Simply put, a labeled node refers to a node that has certain attributes. If the label attribute of node $A$ is "$a$", it means that only signals with the label "$a$" can be received and forwarded by node $A$, thereby reducing network resource occupation and information dissemination range. In communication networks, labels can be used to label the types of signals that a node can receive and send.

**Definition 4 (Time-varying arc)** (*Huang, Xu & Zhu, 2022*): An arc $e = (u, v)$ is defined as a time-varying arc if and only if its length $len_e(t)$ is a time-varying function.

In communication networks, time-varying arcs are employed to depict the varying time required for the same data to complete transmission at different time periods over the same communication connection. This variability in transmission time can be attributed to factors such as network congestion, leading to delays in data transmission. The use of time-varying arcs allows for a more nuanced representation of the dynamic nature of data transmission in communication networks.

**Definition 5 (Time-varying network)** (*Huang, Xu & Zhu, 2022*): A directed network $G(V, E, TW)$ is defined as a time-varying network if and only if there is at least one time-varying arc, where the $V$ is the set of nodes, the $E$ is the set of arcs, the $TW$ is the set of time windows of nodes.

**Definition 6 (Time-varying path)**: A path $P(V_P, E_P, L_P)$ is defined as a time-varying path if and only if $\alpha_i + \omega_i = \tau_i$. Where, the $V_P$ is the set of nodes on path $P$; the $E_P$ is the set of arcs on path $P$; and the $L_P$ is the set of label of nodes on path; the $\alpha_i$ and $\tau_i$ are the arrival time and departure time of $i$th node on the path, respectively; and $\omega_i \geq 0$ is the waiting time at $i$th node.

For any time-varying path $P(V_P, E_P, L_P)$, where the $V_P = \{v_1, v_2, \dots, v_{n+1}\}$, and the $E_P = \{e_1, e_2, \dots, e_n\}$, the $L_P = L_{v_1} \cup L_{v_2} \cup \dots \cup L_{v_{n+1}}$, the length of path $P$ is equal to $len_P = \sum_{i=1}^{n} \left( d_{e_i}(\tau_i) + \omega_i \right) = \alpha_{n+1} - \tau_1$.

**Definition 7 (Label-constrained time-varying shortest route query problem, LTSRQ):** Given a time-varying network $G$, a LTSRQ $Q = (s, z, t_s, L^c)$ is to find a time-varying path $P$ from $s$ to $z$, such that: 1) the $L_P \in L^c$; 2) the $len_P \leq len_{P'}$. Where, the $s$ is source node, the $z$ is destination node, the $L^c$ is the constrained label set, the $P'$ is any satisfied label-constrained path from node $s$ to node $z$ on network $G$. Its mathematical model is:

$$\min \sum_{i \in V, (i,j) \in E} x_i \cdot len_{i,j}(t)$$

$$\text{s.t.} \quad \sum_{j=1, (1,j) \in E}^{n} x_{1,j} - \sum_{j=1, (j,1) \in E}^{n} x_{j,1} = 1$$

$$\sum_{j=1, (n,j) \in E}^{n} x_{n,j} - \sum_{j=1, (j,n) \in E}^{n} x_{j,n} = -1$$ (1)

$$\sum_{j=1, (i,j) \in E}^{n} x_{i,j} - \sum_{j=1, (j,i) \in E}^{n} x_{j,i} = 0, i \neq 1, i \neq n$$

$$x_{i,j} = 0, (i,j) \in E$$

$$l \in L_i, \forall l \in L^c$$

## WDNN ARCHITECTURE

In this section, the architecture of the proposed WDNN is initially presented, followed by the introduction of a WDNN algorithm for addressing the shortest route problem within the context of time-varying network label constraints. furthermore, two theorems is provided to analyze the time complexity and correctness of the proposed algorithm.

### Design of WDNN

The wave delay neuron network is an auto wave neuron-based neural network. Using WDNN to address LTSRQ, the structure of WDNN depends on the topology of the time-varying network, *i.e.,* each node and arc on the time-varying network respectively correspond to a neuron and a link (synapse) that between two neurons. The operating mechanism of the wave delay neural network is as follows: first, activate the root neuron. For non-root neurons, they will only be activated after receiving valid waves (waves that comply with their own label constraints); only activated neurons can generate concurrent waves; the neural network stops running when it reaches the given delay threshold, and the destination neuron selects the shortest route among all the received waves, which is the label-constrained time-varying shortest route.

Auto wave is the medium for neurons to transmit information, which also is regarded as the data packet. As a data packet transmission on an arc, there are delay and cost associated with a wave travel the corresponding synapse, where the delay is calculated by the synapse and the label is calculated by the neuron that sent the wave. Each wave contains three information, namely $P_{g,i}^t$, $A_{g,i}^t$, and $L_{g,i}^t$.

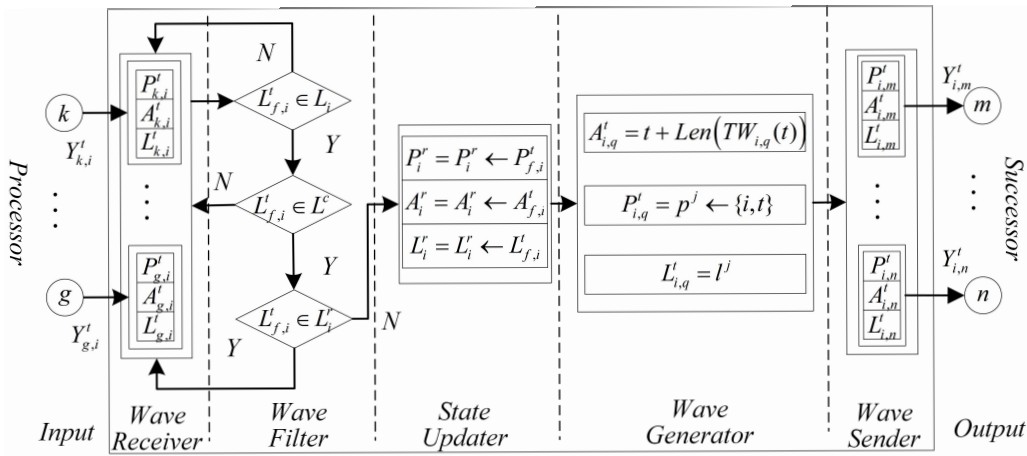

**Figure 1** The structure of a general neuron on WDNN.

Figure 1 shows a general auto wave neuron's structure. Each auto wave neuron consists of seven parts: input, wave receiver, wave filter, state updater, wave generator, wave sender, and output. The illustration and function of each part as following:

1. *Input*: The input of neurons is usually composed of multiple ports used to receive waves sent by other neurons. The number of input ports often depends on the in-degree of the neuron.

2. *Wave receiver*: The wave receiver is used to receive, cache, and decode auto waves. The wave receiver layer consists of several sub receivers, whose number depends on the number of input ports, which also enables each input port to correspond to one sub receiver one by one. When a neuron receives a wave at the current moment, $P_{g,i}^t$, $A_{g,i}^t$, and $L_{g,i}^t$ in its corresponding sub receivers will be assigned based on the information of the wave; if no waves are received, then $P_{g,i}^t$, $A_{g,i}^t$, and $L_{g,i}^t$ will be assigned an initial value. Where, the $P_{g,i}^t$ is used to cache the path in the wave sent by neuron $g$ to current neuron $i$, the $A_{g,i}^t$ is used to cache the arrival time of the wave, and the $L_{g,i}^t$ is used to cache the labels in the wave.

$$P_{g,i}^t = \begin{cases} P_{g,i}^t, & \text{Receive a wave } Y_{g,i}^t \text{ at time } t. \\ null, & \text{Not receive a wave at time } t. \end{cases} \quad (2)$$

$$A_{g,i}^t = \begin{cases} A_{g,i}^t, & \text{Receive a wave } Y_{g,i}^t \text{ at time } t. \\ M, & \text{Not receive a wave at time } t. \end{cases} \quad (3)$$

$$L_{g,i}^t = \begin{cases} L_{g,i}^t, & \text{Receive a wave } Y_{g,i}^t \text{ at time } t. \\ null, & \text{Not receive a wave at time } t. \end{cases} \quad (4)$$

3. *Wave filter*: Wave filters are used to filter the data in the wave receiver. Firstly, based on the label information of the wave, select the wave that the current neuron can process, next determine whether the wave type meet the constrained label, and then determine whether the type of wave has been received. If the wave type is not a type that the current neuron can recognize or not meet the label constrain or has already received

the type of wave, so the wave will be abandoned (since the length of the first received wave must be the shortest, only the earliest arriving wave needs to be recorded.).

4. *State updater*: The state updater is used to update and record the latest state of the current neuron. It includes three sub modules: $P_i^r$, $A_i^r$ and $L_i^r$, which are used to update and record the current shortest route sequence, the arrival time of the wave, and the label of the received wave.

5. *Wave generator*: The wave generator is used to calculate the values of new auto waves. It consists of three parts: $P_{i,q}^t$, $A_{i,q}^t$, and $L_{i,q}^t$, $q \in V_i^F$, their expressions are as following:

$$\begin{cases} P_{i,q}^t = p_j \leftarrow \{i, t\} \\ A_{i,q}^t = t + len(TW_{i,q}(t)) \\ L_{i,q}^t = l^j \end{cases} \tag{5}$$

where, the $l^j \in L_i^r$ is the one of label momerized by current neuron.

6. *Wave sender*: The wave sender is used to encode and send waves, which may be regarded as the inverse process of the wave receiver. It consists of $P_{i,q}^t$, $A_{i,q}^t$, and $L_{i,q}^t$, $q \in V_i^F$.

7. *Output*: The output is the port of auto wave output to successor neurons. Its function is similar to the axon site of biological neurons. The number of output ports depends on the current neuron output.

## WDNN algorithm

The underlying idea of using WDNN to solve LTSP according to the following mechanisms: (1) initialize all neurons and activate the root neuron; (2) all non-root neurons receive auto waves, update neuron's state at special time step; (3) all activated neurons generate auto waves and send to its successor neurons at special time step; (4) the shortest path depends on the wave that arrive destination neuron earliest and satisfied the label constrain $L^c$. Note that, the condition for activate non-root neuron is that the wave receiver receives one or more waves. The detailed procedures of the WDNN algorithm are summarized as shown in Algorithm 1–3. All symbols that used in Algorithm 1–3 are summarized in Table 1.

*Algorithm 1*
*WDNN*
*Input*: $V$, $E$, $L$, $s$, $d$, $\Delta t$, $k$, $L^c$;
*Output*: report label-constrained shortest route;

1: $t = t_s$; /*Initialize neuron timer.*/
2: initializing each neuron by using INA;
3: **while** $L_d^r == \emptyset$ *and* $t - t_s <= k$ **do**
4:     update each neuron by using UNA;
5:     $t = t + \Delta t$; /*Iterative update of neuron timer.*/
6: **end while**
7: report the shortest route $P_d^t$.

*Algorithm 2*
*Initializing neuron algorithm (INA)*
*Input*: $i$, $d$, $t$;
*Output*: $P_i^r$, $A_i^r$, $L_i^r$;

1: **if** $(i = r)$ **then** /* Initializing root neuron */
2:     set $P_i^r = P_i^r \leftarrow i$;
3:     set $A_i^r = A_i^r \leftarrow t$;
4:     set $L_i^r = L_i$;
5: **end if**
6: **if** $(i \neq d)$ **then** /* Initializing non-root neuron */
7:     set $P_i^r = \emptyset$;
8:     set $A_i^r = \emptyset$;
9:     set $L_i^r = \emptyset$;
10: **end if**

*Algorithm 3*
*Updating neuron algorithm (UNA)*
*Input*: $i, L_i, L^c, L_i^r, t, V_i^F, V_i^P, Y_{f,i}^t$; /* $f \in V_i^P$.*/
*Output*: $Y_{i,q}^t$; /* $q \in V_i^F$. */;

1: **for** $f \in V_i^P$ **do** /*Receive waves sent by precursor neurons.*/
2:     **if** $Y_{g,i}^t \neq \emptyset$ **then**
3:         set $P_{f,i}^t = P_{f,i}^t \in Y_{f,i}^t$;
4:         set $A_{f,i}^t = A_{f,i}^t \in Y_{f,i}^t$;
5:         set $L_{f,i}^t = L_{f,i}^t \in Y_{f,i}^t$;
6:     **else**/*No wave received, set receiver to initial value.*/
7:         set $P_{f,i}^t = \emptyset$;
8:         set $A_{f,i}^t = M$;
9:         set $L_{f,i}^t = \emptyset$;
10:     **end if**
11:     **if** $L_{f,i}^t \in L_i$ *and* $L_{f,i}^t \in L^c$ **then** /*Determine whether the received wave satisfies the label constraints of the current neuron and whether this type of wave has been received.*/
12:         **if** not $L_{f,i}^t \in L_I^t$ **then**
13:             $P_i^r = P_i^r \leftarrow P_{f,i}^t$;
14:             $A_i^r = A_i^r \leftarrow A_{f,i}^t$;
15:             $L_i^r = L_i^r \leftarrow L_{f,i}^t$;
16:         **end if**
17:     **end if**
18: **end for**
19: **for** $j \in V_i^F$ **do** /*Send waves to each succeeding neuron.*/
20:     set $A_{i,q}^t = t + len(TW_{i,q}(t))$;
21:     set $P_{i,q}^t = p^j \leftarrow \{i, t\}$;
22:     set $L_{i,q}^t = l^j$;
23:     set $Y_{i,q}^t = \{P_{i,q}^t, A_{i,q}^t, L_{i,q}^t\}$;
24: **end for**

## Time complexity of WDNN

*Theorem 1.* Let $n$ be the number of nodes on the time-varying network, the $m$ is the number of all arcs, the $V_i^P$ be the number of the neuron $i$'s input arcs, the $V_i^F$ be the number of the neuron $i$'s output arcs, $k$ be the arrival time of destination node on output path, and $\Delta t$ is the step (unit) of iteration. The time complexity of WDNN is equal to $O\left(\frac{2k}{\Delta t} \cdot m + n\right)$.

*Proof:* The WDNN algorithm consists of four main steps (step 1: line 1; step 2: line 2; step 3: line 3-6; step 4: line 7), the time complexity of step 1 and step 4 are all equal to $O(1)$ due to without loop, iteration or recursion. The step 2 and step 3 are relatively complicated operations, the detailed analysis as following:

As to step 2 in WDNN, all neurons need to call INA for initializing. The times for running INA depends on the number of neurons in the neural network. Furthermore, the INA does not contain loop. Then, the time complexity of this step is equal to $O(n)$.

Step 3 in WDNN is a loop, the number of iterations of the loop is limited by the $k$. Then, each neuron needs to run UNA for update at each time, which times depends on the number of neurons on the neural network. As to UNA, each neuron needs to send a wave to its precurssors and successors, its complexity is determined by $V_i^P + V_i^F$. Therefore, the time complexity of this step is equal to $O\left((k/\Delta t) \cdot \sum_{i=1}^{n} V_i^P + V_i^F\right)$.

In summary, the time complexity of the WDNN algorithm is equal to:

$$O\left(n + \frac{k}{\Delta t} \cdot \sum_{i=1}^{n} m_i\right) = O\left(\sum_{i=1}^{n} \left(1 + \frac{k}{\Delta t} \cdot \left(V_i^P + V_i^F\right)\right)\right) = O\left(\frac{2k}{\Delta t} \cdot m + n\right). \tag{6}$$

It is worth noting that WDNN is a parallel algorithm, all neurons on the neural network are calculated in parallel. Therefore, in an ideal situation, the number of neurons does not affect the algorithm execution speed, the theoretical time complexity of WDNN algorithm is equal to $O\left(1 + \frac{k}{\Delta t} \cdot \left(V_i^P + V_i^F\right)\right)$.

## Correctness of WDNN

*Theorem 2.* The first auto-wave that arrives at the destination neuron and satisfies the label constraint determines the shortest route from root neuron to destination neuron.

*Proof:* Let $x_1$, $x_2$, and $x_3$ be the precursor neurons of neuron $z$ (see Fig. 2). If the first auto-wave that received by neuron $z$ is sent by neuron $x_1$, then the delay is $T_{P_1} + w_{x_1} + T_{P_2}$, the label set is $\{a, b, c\}$. If the second auto-wave that received by neuron $z$ is sent by neuron $x_2$, then the delay is $T_{P_3} + w_{x_3} + T_{P_4}$, the label set is $\{a, b\}$. If the third auto-wave that received by neuron $z$ is sent by neuron $x_3$, then the delay is $T_{P_5} + w_{x_3} + T_{P_6}$, the label set is $\{a, b\}$. Because the destination neuron $z$ will no longer receive the auto-wave after receiving the auto-wave that meets the label threshold, so if the second automatic wave is received, it is apparent that the label $\{c\}$ is not in the constrained label set; if the third auto-wave is received by neuron $z$, it is apparent that $T_{P_3} + w_{x_3} + T_{P_4} > T_{P_5} + w_{x_3} + T_{P_6}$, in reality, it contradicts the algorithmic process. In summary, *Theorem 2* is correct.

## EXPERIMENTAL RESULTS AND DISCUSSION

To evaluate the performance of the proposed algorithm, the performance of WDNN is compared with the well-known algorithm of *Yang & Zhou (2017)* (Yang), the algorithm of

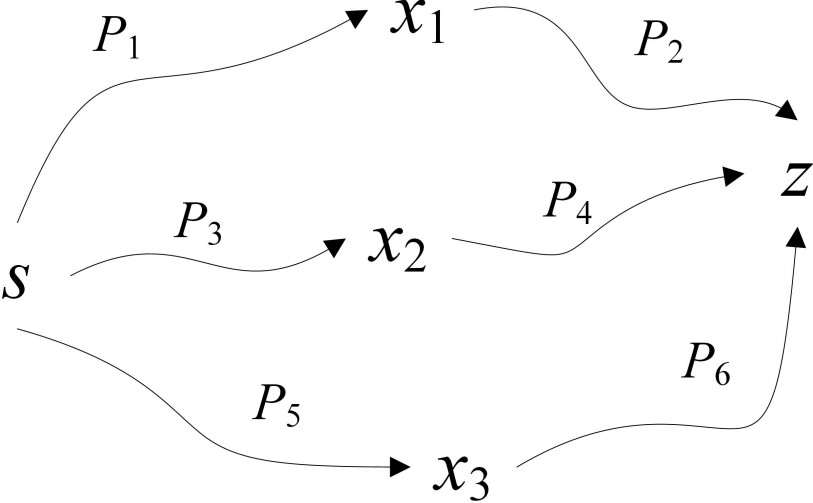

**Figure 2** Proof of Theorem 2.

**Table 2** The structure of each dataset.

| Dataset | Number of nodes | Number of edges | Number of time-windows | Length of edge |
|---------|-----------------|-----------------|------------------------|----------------|
| 50 | 50 | 400 | [1,5] | [1,20] |
| 100 | 100 | 800 | [1,5] | [1,20] |
| 150 | 150 | 1,200 | [1,5] | [1,20] |
| 200 | 200 | 1,600 | [1,5] | [1,20] |
| N-Net | 4,941 | 13,203 | [1,5] | [1,20] |
| I-Net | 22,962 | 96,872 | [1,5] | [1,20] |

**Table 3** Relative error of algorithms on datasets with different nodes.

| Algorithm | Number of nodes | | | |
|-----------|-----|-----|-----|-----|
| | **50** | **100** | **150** | **200** |
| Veneti | 0.383 | 0.131 | 0.283 | 0.237 |
| Yang | 0.191 | 0.219 | 0.123 | 0.201 |
| Tu | 0.019 | 0.025 | 0.026 | 0.027 |
| WDNN | 0.000 | 0.000 | 0.000 | 0.000 |

*Veneti, Konstantopoulos & Pantziou (2015)* (Veneti), algorithm of *Tu et al. (2020)* (Tu) on 120 randomly generated label time-varying networks using public network generation tools *Random* with different number of nodes and on two public real dataset neural network (N-Net) and Internet Network (I-Net) (https://www.diag.uniroma1.it/challenge9/download. shtml). The structure of each dataset is shown in Table 2. The space complexity of WDNN, Veneti, Yang and Tu are $O((k/\Delta t)\cdot n)$, $O((k/\Delta t)\cdot n)$, $O(n)$ and $O(n\cdot e)$, respectively; the time complexity of WDNN, Veneti, Yang and Tu respectively is $O\left(\frac{2k}{\Delta t}\cdot m+n\right)$, $O((k/\Delta t)\cdot(n+m))$, $O(n^2)$ and $O(n^2)$.

The performance of proposed algorithm are evaluated from two aspects: number of nodes and number of time windows. In all experiments, without loss of generality, each experiment will be conducted $N = 20$ times, and the source and destination nodes will be randomly selected in each repeated experiment. All programs and instances running a machine with Intel Xeon(R) Gold 5218R CPU and 64G RAM, and all programs are implemented in C#.

For convenience, the relative error (RE) as an index to compare the performance of Yang, Veneti, Tu, and WDNN. The calculate expression of RE is as following:

$$RE = \sum_{i=1}^{N} \left( \frac{|C_i^V - O_i^V|}{O_i^V} \right) / N \tag{7}$$

where the $C_i^V$ is the calculated value of $i$th experiment, and the $O_i^V$ is the optimal value of $i$th experiment.

### Effect of different nodes

In this experiment, the performance of the proposed algorithm is evaluated by varying number of nodes between 50 to 200. Table 3 shows the effectiveness of the proposed algorithm and existing algorithms in solving 40 randomly generated label time-varying networks with different nodes. As shown in Table 3, compared to Yang, Veneti and Tu, the proposed algorithm obtain the optimal solution of the problem, while Yang algorithm has a relative error ratio between 0.123 and 0.219, the Veneti algorithm has a relative error ratio between 0.131 and 0.383, and the relative error ratio of Tu algorithm is shown a increasing trend from 0.019 to 0.027. It can be seen that the change in the number of nodes does not affect the accuracy of the Veneti, Yang and WDNN algorithms. This is because changes in the number of nodes only cause changes in the network size, while the degree and edge length between nodes do not have any significant changes, as the number of nodes does not affect the accuracy of the three algorithm. However, as the network size increases (the number of nodes increases), the error ratio of Tu algorithm is showing an upward trend, which means that Tu is not suitable for label-constrained shortest route solving on large time-varying networks. Furthermore, the reason why the algorithm proposed in this article can obtain the optimal solution on label time-varying networks with different number of nodes (network size) is that the neural network maps each node to a neuron, and changes in network size only cause changes in the network size, that is, an increase in the number of neurons, so it does not affect the performance of the algorithm. The compute time with different nodes are shown in Fig. 3.

In terms of computational time, although the proposed algorithm has a slightly slower computational speed than Yang algorithm when the network size is small (between 50 and 150 nodes), the loudness speed of WDNN is actually better than Yang, Veneti and Tu algorithms when the network size is large. It is because that the Yang algorithm adopts a heuristic search mechanism similar to the Dijkstra algorithm, which does not require synchronization in the time dimension. On large scale networks, the advantages of the proposed algorithm are presented due to the parallel computation of each neuron. The Veneti and Tu algorithms requires a lot of computation time due to the need to handle

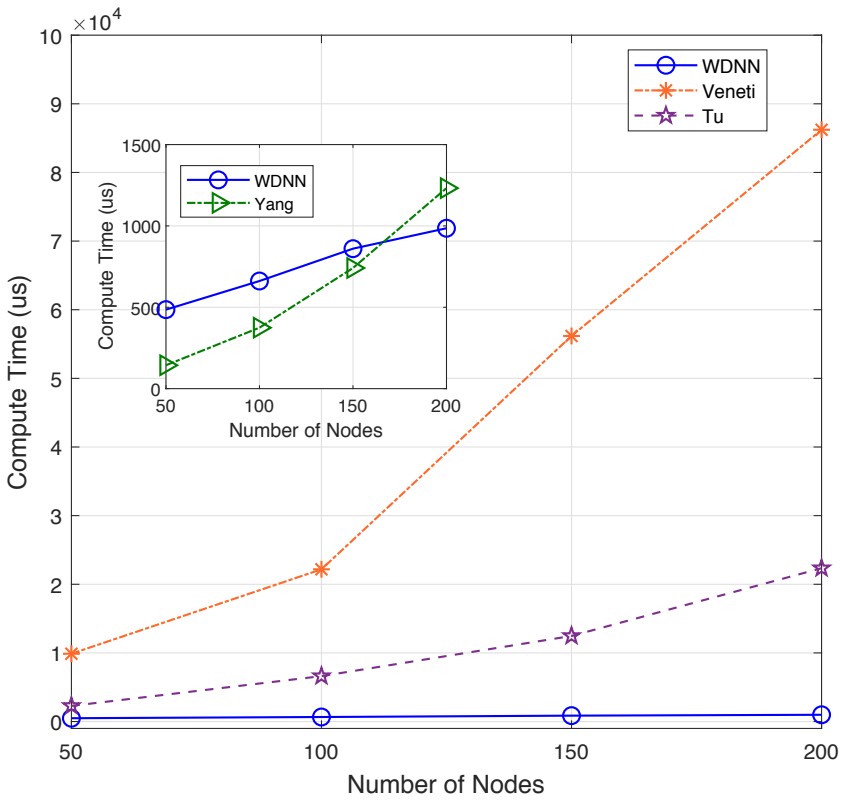

**Figure 3** **The compute time with different nodes.**

a large scale number of labels. In summary, although the proposed algorithm is slightly slower than Yang algorithm on smaller networks, it has better solution accuracy. On larger networks, the proposed WDNN outperforms existing algorithms in terms of response speed and solution accuracy.

## Effect of different time windows

In this experiment, the performance of the proposed algorithm is evaluated by varying number of time windows between 1 to 5. Table 4 shows the effectiveness of the proposed algorithm and existing algorithms in solving 50 randomly generated label time-varying networks with different time windows. As shown in Table 4, compared to Yang, Veneti and Tu, the proposed algorithm obtain the optimal solution of the problem, while Yang algorithm has a relative error ratio of 0.15 to 0.22, the Veneti algorithm has a relative error ratio of 0.12 to 0.38, and the relative error ratio of Tu from 0 to 0.028. Figure 4 shows the relative error trend of the four algorithms when the number of time windows for each arc changes from 1 to 5. From Fig. 4, it can be seen that the proposed algorithm can obtain the optimal solution on time-varying networks with different number of time windows. The relative error of Yang and Tu algorithms increases with the increase of the number of time windows. Although the relative error of Veneti algorithm shows a decreasing trend, there

**Table 4  Relative error of algorithms on datasets with different time windows.**

| Algorithm | Number of time windows | | | | |
|---|---|---|---|---|---|
| | 1 | 2 | 3 | 4 | 5 |
| Veneti | 0.380 | 0.270 | 0.236 | 0.150 | 0.120 |
| Yang | 0.158 | 0.160 | 0.157 | 0.188 | 0.211 |
| Tu | 0.000 | 0.021 | 0.023 | 0.025 | 0.028 |
| WDNN | 0.000 | 0.000 | 0.000 | 0.000 | 0.000 |

is still an error of over 0.1 at 5 time windows. Figure 5 shows the compute time trend of the proposed WDNN, Yang, Veneti and Tu algorithms on a network with varying number of time windows. As shown in Fig. 5, both WDNN, the Yang and Tu algorithms show an upward trend with the increase of the number of time windows. This is because as the number of time windows increases, the algorithm needs to consume a certain amount of time when selecting a time window. Although the query time of the Veneti algorithm does not show an upward trend, this is because the time spent selecting the time window is relatively small compared to the search path of the Veneti algorithm, so it is not shown. Furthermore, the speed at which the proposed algorithm increases with the number of time windows is smaller than that of the Yang and Tu algorithms, while the Veneti algorithm has a computation time that is one order of magnitude higher than the proposed algorithm. In the case of more time windows, the proposed algorithm still has the best performance. In summary, the proposed algorithm has better performance compared to existing algorithms with varying time windows.

## Experimental results on large-scale networks

This experiment will evaluate the performance of the proposed algorithm on large-scale real-world networks. Tables 5 and 6 show the response times of the proposed WDNN algorithm and Veneti, Yang, and Tu algorithms on real networks N-Net and I-Net, respectively, for solving the time-varying label-constrained shortest route problem on subnets with different number of time windows. Meanwhile, Figs. 6 and 7 respectively show the relative errors of the WDNN algorithm and Veneti, Yang, and Tu algorithms in solving the time-varying label-constrained shortest route problem on subnets with different number of time windows in these two real networks. From Table 5, it is evident that in the N-Net network with approximately 4,000 nodes, the proposed algorithm shows a significant improvement in computational speed compared to Veneti and Yang algorithms. Furthermore, compared to Tu algorithm, the computational speed of WDNN has also increased by about twice. In an I-Net network with approximately 20,000 nodes, it can be clearly observed from Table 6 that the proposed algorithm shows a significant improvement in computational speed compared to Veneti, Yang, and Tu algorithms. This result indicates that the proposed algorithm is better suited for label-constrained time-varying shortest routing query problems on large-scale networks. Through the comprehensive analysis of Figs. 6 and 7, it can be concluded that the proposed WDNN does not decrease accuracy as the number of time windows increases, and always maintains the ability to query the

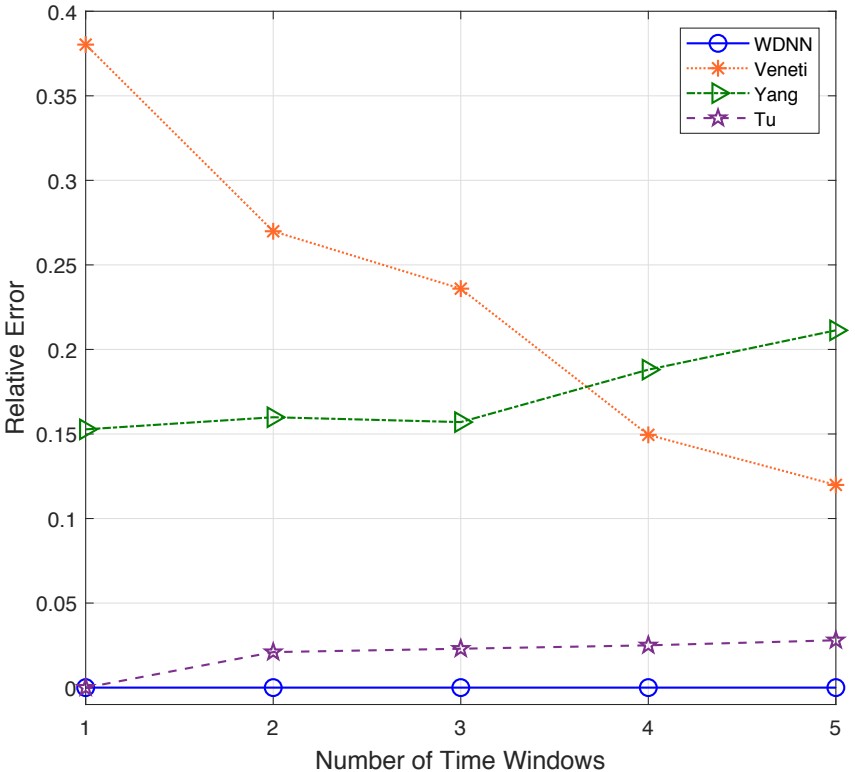

**Figure 4** The relative error with different time windows.

**Table 5** The compute time (ms) with different time windows for the N-Net dataset.

| Algorithm | Number of time windows | | | | |
|---|---|---|---|---|---|
| | 1 | 2 | 3 | 4 | 5 |
| Veneti | 4011.81 | 4567.63 | 4382.80 | 4534.41 | 4582.48 |
| Yang | 69259.23 | 73693.86 | 62957.29 | 71008.40 | 65617.65 |
| Tu | 334.51 | 416.62 | 353.41 | 391.93 | 360.54 |
| WDNN | 172.59 | 169.16 | 158.19 | 161.26 | 162.71 |

optimal solution. This is because WDNN is able to flexibly choose the most suitable departure time based on the time window to ensure earlier arrival at the next node. However, other algorithms lack a time window selection mechanism, and as the number of time windows increases, the query error shows an upward trend.

## CONCLUSION

In this study, a framework for solving the label-constrained time-varying routing query (LTSRQ) on time-varying networks is proposed using a wave delay neural network (WDNN). WDNN is comprised of self-designed seven-layer auto wave neurons, enabling parallel computing. Unlike other intelligent or neural network algorithms, the proposed neural network operates as an intelligent algorithm without the need for training. This
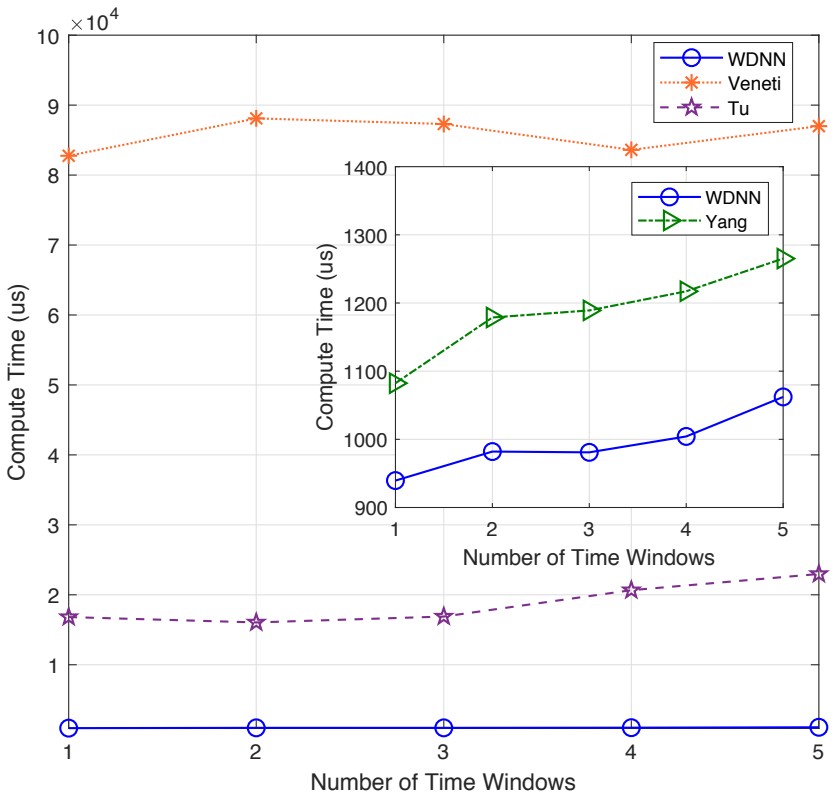

**Figure 5** The compute time with different time windows.

**Table 6** The compute time (ms) with different time windows for the I-Net dataset.

| Algorithm | Number of time windows | | | | |
|---|---|---|---|---|---|
| | 1 | 2 | 3 | 4 | 5 |
| Veneti | 26789.26 | 27723.212 | 27076.32 | 26779.87 | 25608.23 |
| Yang | 22011.51 | 22287.60 | 21343.86 | 20870.71 | 20851.77 |
| Tu | 28820.42 | 30466.24 | 26013.79 | 22377.93 | 23775.95 |
| WDNN | 1516.66 | 1424.55 | 1585.39 | 1431.50 | 1250.00 |

mitigates the issue of slow response speed associated with training, diminishing the impact of network size (number of nodes) on model performance and considerably expediting the solution process on complex networks. In comparison to existing algorithms, the proposed WDNN demonstrates the capability to obtain the global optimal solution and provides interpretability. Through experiments conducted on 120 time-varying networks with varying node numbers and time windows randomly generated using the public network generation tool Random, as well as on real networks N-Net and I-Net, it is observed that

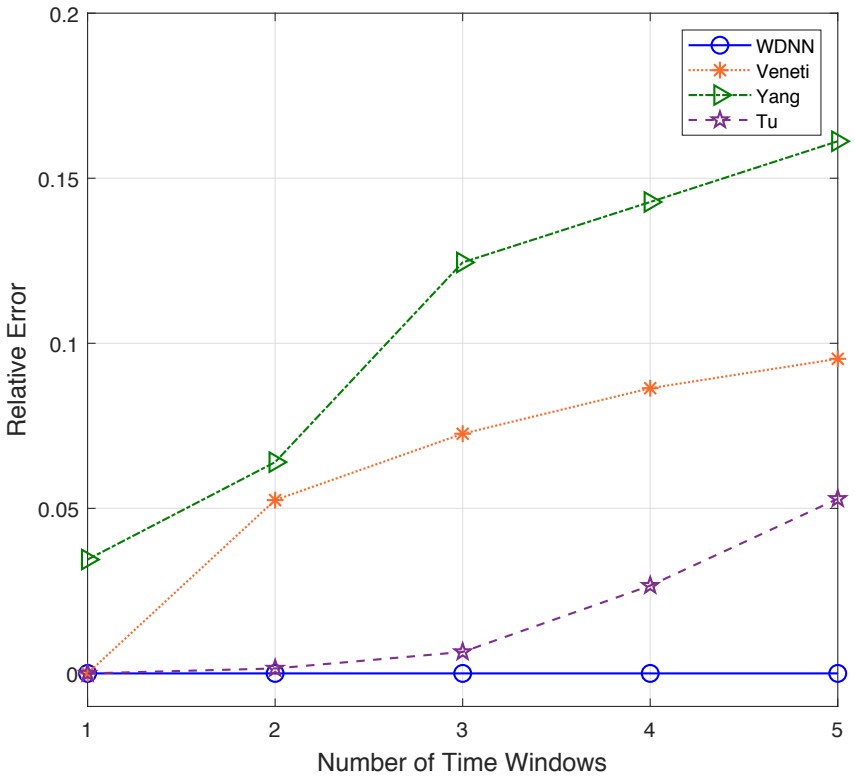

**Figure 6** **The relative error with different time windows for the N-Net dataset.**

the WDNN outperforms existing algorithms such as Veneti, Yang, and Tu. This offers substantial evidence for the effectiveness of WDNN in addressing the LTSRQ problem. In practical applications, multiple uncertain properties often characterize networks, and the label-constrained shortest route query problem on time-varying networks in uncertain environments has not been addressed by the proposed WDNN. In future work, attention should be directed towards improving the structure of neural networks or neurons to enhance algorithm adaptability in uncertain and time-varying environments, including aspects of fuzziness and randomness. When enhancing neurons, the primary focus should be on refining their wave filters, state updates, and wave generators. Wave filters play a crucial role in determining the efficiency of pathfinding, while state updates and wave generators influence the accuracy of pathfinding. For fuzzy time-varying environments, the addition of fuzzy simulation units is recommended to handle fuzzy edge lengths. In the case of randomly time-varying environments, incorporating a random simulation unit is advisable to calculate the probability distribution of the path. These enhancements will contribute to the overall robustness and applicability of the proposed WDNN in handling uncertainties within network environments.

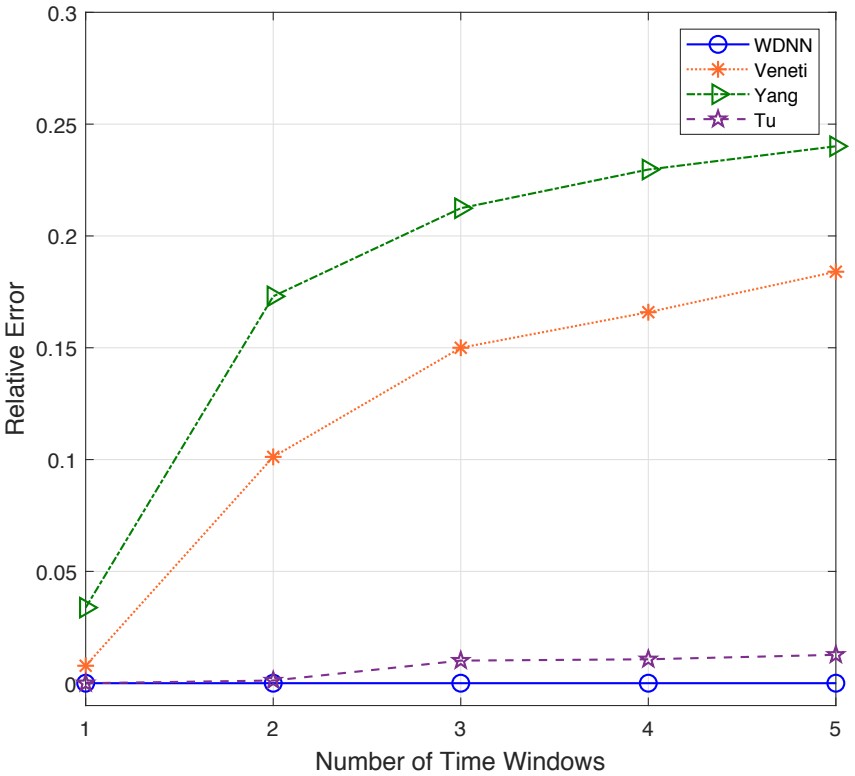

**Figure 7   The relative error with different time windows for the I-Net dataset.**

### Funding
This research was funded by the National Key R&D Plan Program grant number 2022YFF0609602, the State Administration for Market Regulation Science and Technology Plan Project grant number 2022MK187, and the President Funding Project of China National Institute of Standardization grant number 242023Y-10413. The funders had no role in study design, data collection and analysis, decision to publish, or preparation of the manuscript.

### Grant Disclosures
The following grant information was disclosed by the authors:
National Key R&D Plan Program: 2022YFF0609602.
The State Administration for Market Regulation Science and Technology Plan Project: 2022MK187.
The President Funding Project of China National Institute of Standardization: 242023Y-10413.

### Competing Interests
The authors declare there are no competing interests.

## Author Contributions

- Bing Han conceived and designed the experiments, performed the experiments, analyzed the data, performed the computation work, prepared figures and/or tables, authored or reviewed drafts of the article, and approved the final draft.
- Qiang Fu conceived and designed the experiments, performed the experiments, analyzed the data, performed the computation work, authored or reviewed drafts of the article, and approved the final draft.
- Xinliang Zhang performed the experiments, analyzed the data, performed the computation work, prepared figures and/or tables, and approved the final draft.

## Data Availability

The code and data is available at Zenodo:

Fu, Q. (2023). LTSRQ. Zenodo. https://doi.org/10.5281/zenodo.10443736

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
