# Peer review of "A wave delay neural network for solving label-constrained shortest route query on time-varying communication networks"

_PeerJ Computer Science, doi:10.7717/peerj-cs.2116_

## Round 0.1 · original submission · Major Revisions

Your manuscript entitled " A wave delay neural network for solving label-constrained shortest route query on time-varying communication networks" has been reviewed. Reviewers felt that more works, in particular, description of the data set and comparison of the proposed method with others should be clarified, are required. Hence, a "Major Revision" is recommended.

Reviewer 4 has suggested that you cite specific references. You are welcome to add it/them if you believe they are relevant. However, you are not required to include these citations, and if you do not include them, this will not influence my decision.

**Language Note:** The review process has identified that the English language must be improved. PeerJ can provide language editing services - please contact us at copyediting@peerj.com for pricing (be sure to provide your manuscript number and title). Alternatively, you should make your own arrangements to improve the language quality and provide details in your response letter. – PeerJ Staff

Reviewer 1 ·

Basic reporting

In this paper, a label-constrained time-varying shortest route query problem (LTSRQ) was introduced, The purpose of LTSRQ is to find a route from the source node to the destination node having the shortest delay. Based on this paper, the following comments can be founded.
The proposal is appealing and engaging, and the research deserves consideration, but I'm not sure about the novelty and contributions of this article because related researches can be found in the literature. Therefore, the authors have to make an effort to explain with more detail the main contributions of this paper within the domain of the research (differing from previous works).

Experimental design

So many symbols are employed in this paper, thus, the definition of the symbol should be given in the section of the Introduction. Especially in some definitions, the symbol is very hard to follow.

It is hard for readers to understand what is the label-constrained time-varying shortest route query problem. Why not give more description in the Introduction.
In the WDNN ARCHITECTURE, a wave delay neural network model is proposed, while, readers cannot be understood easily, thus, why not give a concrete neural network expression.
In the Design of WDNN, “Input: The input of neurons is usually composed of multiple ports used to receive waves sent by other neurons. The number of input ports often depends on the degree of penetration of the neuron.”, what is this penetration?

Validity of the findings

The proposal is appealing and engaging, and the research deserves consideration, but I'm not sure about the novelty and contributions of this article because related researches can be found in the literature.

Additional comments

For the Pseudocode of the proposed model algorithm, please add appropriate instructions to facilitate readers' understanding.
In Definition 7, the format of Equation (1) is not aesthetically pleasing and requires further standardization.
The writing in this manuscript is too colloquial.

Reviewer 2 ·

Basic reporting

In this paper, a wave delay neural network (WDNN) is proposed to address a label-constrained shortest route query problem and the theoretical analysis of the proposed algorithm is carried out from two aspects of time complexity and correctness. I have some minor concerns:
1.What is the advantages of your proposed WDNN compared to other scheme? Is there any disadvantage?
2.Please highlight the novelty and contribution in the introduction.
3.Please check the English of the whole paper.
4.Please update the references.
5.In subsection “Correctness of WDNN” and figure 2, where is theorem 3?
6.Please elaborate further on the future work.

Experimental design

N/A.

Validity of the findings

N/A.

Additional comments

N/A.

Reviewer 3 ·

Basic reporting

The presented paper proposed a wave delay neural network to solve label-constrained shortest route query problem on time-varying communication networks. The preliminaries are well-presented and structure of the paper is well-organized. However, there are several issues may need further consideration. 

1)The problem to solve is not convincing as important in communication network. In the introduction section, it is recommended to present thoroughly.

2)How do you move from discussing shortest route query problem to this paper?

3)Why is solving label-constrained shortest route query problem important in the area of study?

4) Why do you choose to compare with those methods? It is recommended to better illustrate the compared baseline methods.

5)There are conflicts in the application of symbols in the manuscript. For example, the destination node is represented by d and z respectively. It is recommended to normalize it.

6)The writing of the manuscript uses too much "we". It is recommended to adjust the expression to make it more written.

There are irregularities in the references, please check and revise them carefully.

Experimental design

NONE

Validity of the findings

NONE

Additional comments

NONE

Reviewer 4 ·

Basic reporting

1) The references of all definitions should be added.
2) The converegence and stability of the solution should be investigated.
3) Some related wave and neural networks works should be added:
-Dynamics on time scales of wave solutions for nonlinear neural networks. Waves in Random and Complex Media, pp.1-17. (2022)
-Almost anti-periodic solution of inertial neural networks model on time scales. In MATEC Web of Conferences (Vol. 355, p. 02006). EDP Sciences. (2022)
-Dynamics of delayed cellular neural networks in the Stepanov pseudo almost automorphic space. Discrete and Continuous Dynamical Systems-S, 15(11), pp.3097-3109. (2022)
-Stability analysis of inertial neural networks: A case of almost anti‐periodic environment. Mathematical Methods in the Applied Sciences, 45(16), pp.10476-10490. (2022)
-Applications of artificial neural network to solve the nonlinear COVID-19 mathematical model based on the dynamics of SIQ. Journal of Taibah University for Science, 16(1), pp.874-884. (2022)
4) How have you validate the model?
5) Have you detected an overfitting problem?

Experimental design

See above

Validity of the findings

See above

Additional comments

See above

Reviewer 5 ·

Basic reporting

The authors propose a wave delay neural network (WDNN) based on auto-wave neurons which all compute in parallel, without training, for solving the label-constrained time-varying shortest route query problem (LTSRQ). The time complexity and correctness of the algorithm are analyzed, and the experimental results show superior performances to existing methods.
The following observations should be addressed:
1. On page 6, the unnumbered theorem should be Theorem 2 and Theorem 2 should be Theorem 3. Please correct this aspect.

Experimental design

2. The proposed algorithm is compared to two algorithms from 2015 and 2017, respectively, which are rather old. Aren’t there other newer algorithms solving the same problem to compare the proposed algorithm with?

Validity of the findings

3. The proposed algorithm is just tested on synthetic, generated data. Is it not possible to test the algorithm on real-world applications?

Additional comments

4. Minor points: “it accurately simulate” should be “it accurately simulates”, “reachibility" should be “reachability”, “in section 5” should be “in Section 4”, “is denotes as” should be “is denoted as”, “the a_{i} and \tau_{i}” should be “the \alpha_{i} and \tau_{i}”, “cosntrained” should be “constrained”, “wave; If” should be “wave; if”, “momeried” should be “memorized”, “neurons.Its” should be “neurons. Its”, “path depend” should be “path depends”, “algorithm 1-3” should be “Algorithm 1-3”, “that arriving” should be “that arrives”, “are implements” should be “are implemented”, “50 to 100” should be “50 and 100”, “dijkstra” should be “Dijkstra”.
5. In general, the article should be reviewed to correct spelling errors, and to improve the English language.

Reviewer 6 ·

Basic reporting

THe contents of this paper is ok. However, there are some contents that should be promoted. They are as follows:

1.There are some mistakes in the English writing of this manuscript, such as in the theorem “Let n be the numbers of nodes”, please check and revise the full text carefully.

2. This paper considers a label-constrained time-varying shortest route query problem on time-varying network with label. The authors propose a neural network method to solve the label-constrained time-varying shortest route query problem. What is the starting point for designing this neural network? The authors should describe in detail the necessity and motivation of the research question in the introduction.

3. In the PRELIMINARIES section, the authors give a series of definitions, but the lack of some necessary explanations makes it difficult for readers to understand these definitions. I suggest that the authors consider adding clarifications to certain definitions.

4. The writing standard of this manuscript needs to be improved, and necessary descriptions should be added between sections and subsections, so that readers can easily understand the main content of this section.

5. In the Design of WDNN section, the authors mainly introduced the structure of the automatic wave neuron, and then the introduction of the structure and operating mechanism of WDNN is not comprehensive enough. Please further explain the structure of the proposed WDNN and how it operates.

Experimental design

The original primary research is suitable to the scope of this journal. However, there are still some problems that are as follows:

1. In the experimental part, the description of the data set used is missing, and I suggest that the authors add a description of the structure of the label time-varying network used.

2.The authors compared the proposed algorithm with the Veneti and Yang algorithms. As far as I know, these two algorithms are not designed for the label-constrained time-varying shortest route query problem. Why choose to compare with these two algorithms?

Validity of the findings

1. In the conclusion part, the research results and main contributions of this paper should be emphasized, including the research questions, the brief content and novelty of the proposed method.

2. For the description of future work, the possible research direction of the problem or algorithm should be described in detail, and in particular, specific ideas on how to improve the proposed method for fuzzy or stochastic uncertain time-varying environments should be given.

---

## Round 0.2 · Minor Revisions

Authors have addressed the majority of the reviewers' comments to their satisfaction. However, it is not clear how these comments and the answers of the comments were implemented in the revised manuscript. Based on this, a minor revision has recommended and authors are encouraged to revise your manuscript taking the consideration of **previous** reviewers' comments.

Reviewer 4 ·

Basic reporting

Yes

Experimental design

Yes

Validity of the findings

Yes

Reviewer 6 ·

Basic reporting

no comment

Experimental design

no comment

Validity of the findings

no comment

Additional comments

The authors made good revisions to address the comments on the previous version of the manuscript. All my questions have been answered and can be accepted.

---

## Round 0.3 · accepted · Accept

Authors have addressed all the comments from the reviewers. This paper is now ready to be published.